# Metabolomic and Transcriptomic Analyses Revealed Lipid Differentiation Mechanisms in *Agaricus bisporus* at Ambient Conditions

**DOI:** 10.3390/jof10080533

**Published:** 2024-07-30

**Authors:** Mengjiao Tao, Yiting Zhu, Faxi Chen, Yilu Fang, Yanqi Han, Guohua Yin, Nanyi Li

**Affiliations:** 1College of Horticulture Science, Zhejiang A&F University, Hangzhou 311300, Chinafaxichen0110@163.com (F.C.);; 2College of Biological and Chemical Engineering, Qilu Institute of Technology, Jinan 250200, China

**Keywords:** *Agaricus bisporus*, metabolomics, transcriptomics, postharvest storage

## Abstract

*Agaricus bisporus* is one of the most popular mushroom species in the world; however, mushrooms are highly susceptible to browning due to the absence of a protective cuticle layer and high respiration rate. The molecular mechanism underlying the process of mushroom browning needs to be explored. Here, we analyzed the transcriptomic and metabolomic data from *A. bisporus* at ambient temperature. Specifically, a total of 263 significantly changed metabolites and 4492 differentially expressed genes were identified. Lipid metabolites associated with cell membrane degradation were predominantly up-regulated during ambient storage. Transcriptomic data further revealed the alterations of the expression of membrane lipid metabolism-related enzymes. Additionally, energy metabolic processes and products such as glycolysis and linoleic acid changed significantly during ambient storage, indicating their potential roles in the quality deterioration of *A. bisporus*. These findings provide new insights into the underlying lipid metabolic mechanisms of *A. bisporus* during postharvest ambient storage and will provide values for mushroom preservation techniques.

## 1. Introduction

*Agaricus bisporus*, the commercially cultivated mushroom species, is a highly popular product worldwide due to its flavor and diverse health benefits, including anticancer, antioxidant, anti-diabetic, antimicrobial, and anti-obesity activity [1,2,3]. Furthermore, numerous studies have highlighted the potential of bioactive compounds derived from *A. bisporus* for developing drugs against serious diseases, as demonstrated through bioactivity analyses [4]. *A. bisporus* lectin (ABL) and mannose-binding protein (Abmb) have been investigated for their potential in medicinal or pharmaceutical applications [5]. China is one of the world’s largest producers of *A. bisporus* mushrooms, with an annual yield exceeding one million tons [6].

The visual appeal of mushrooms is a key determinant in impacting consumer purchasing decisions [7]. It is difficult to conserve the quality of harvested *A. bisporus* due to the absence of a protective cuticle layer and its high respiration rate [8,9]. The quality degradation of a mushroom causes moisture and nutrient loss, browning, tissue softening, and cap opening [10]. In recent years, various postharvest approaches have been implemented to prolong the preservation of *A. bisporus*. For example, physical methods such as modified atmosphere packaging, irradiation, and pulsed electric fields can extend the shelf life of *A. bisporus* [11,12,13]. The senescence of *A. bisporus* is delayed by chemical methods such as antimicrobial agents or electrolytic water cleaning, ozone, and coating treatments [14,15,16,17]. Additionally, the storage temperature is the most crucial factor influencing the respiration, color, and ripening of the mushroom. Increasing the storage temperature accelerates the aging, browning, weight loss, and softening of the mushroom [18]. Therefore, for better results, most postharvest *A. bisporus* preservation methods should be accompanied by low and constant temperature (above 0 °C) [19].

Generally, browning is one of the main causes of quality loss of *A. bisporus* [20]. Browning decreases the commercial value of mushrooms and ultimately leads to considerable economic losses. Substantial evidence indicates that enzymatic browning during storage is the main cause of browning in *A. bisporus* during storage [21]. Enzymatic browning of *A. bisporus* is a complex process involving various phenolic compounds [22], phenoloxidases [23,24], and multiple other enzymes [25,26]. The expression of *ppo* genes in the browning of *A. bisporus* fruit body (pileus, stipe, and gill) was inhibited by UV-C treatment [27]. Moreover, a study integrating metabolomics and transcriptomics reported that *A. bisporus* browning can be affected by the dynamic content of organic acids and trehalose [28]. Maintenance of cell membrane integrity is also an important factor in inhibiting postharvest browning of *A. bisporus*. Lin et al. showed that phase change materials combined with melatonin treatment could regulate the expressions of antioxidant-associated genes and maintain cell membrane integrity to effectively delay the browning of *A. bisporus* [29]. Additionally, melatonin treatments can prominently inhibit electron leakage by significantly increasing the expression levels of *NdufB9* and *RIP1* to protect cell membranes in *A. bisporus* [30].

Significant research advances have been achieved in the preservation of *A. bisporus*. However, the postharvest physiology and metabolism of the mushroom remain uncovered. In recent years, high-throughput techniques have become widely utilized for investigating the molecular-level biological processes and metabolic mechanisms in plants [31,32]. In this study, we utilized metabolomic and transcriptomic data to analyze the dynamics of major metabolites and the underlying regulatory pathways in *A. bisporus* throughout various storage stages. Our research not only sheds light on the postharvest physiology of *A. bisporus* but also contributes to enhancing post-harvest storage practices and the development of effective preservation technologies.

## 2. Materials and Methods

### 2.1. Mushroom Materials and Treatments

White mushrooms (*A. bisporus*, A15) were harvested from the Longchen Modern Agriculture Technology Company in Jiaxing City, Zhejiang Province, China. Mushrooms with consistent shape, size, and excellent quality were carefully selected and promptly transported to the laboratory, where they were pre-cooled at 4 °C overnight. The samples were randomly divided into three groups, each group had three biological replicates, and each replicate contained six *A. bisporus* fruiting bodies. Subsequently, the mushrooms were put into boxes and stored at 23 ± 1 °C (room temperature, RT), and the mushroom flesh (0.5 cm thick) was collected from three storage periods of 0 h (CK), 6 h (RT_6), and 24 h (RT_24). Samples were ground into powder with liquid nitrogen and stored frozen at −80 °C until later use.

### 2.2. Measurements of Browning Index, Firmness and Weight Loss

The degree of browning, also referred to as the browning index, was assessed by calculating the ratio of the browned area to the cap surface and scored on a scale of one to five [33]. Firmness was measured using a Stable Micro System (SMS) TA. XTPlus texture analyzer (Stable Micro System, Godalming, UK). For each mushroom, the 2 mm diameter cylindrical probe of the firmness mirror was pressed into the mushroom at a speed of 5 mm/s, and the hardness was defined as the maximum force in the force versus time curve. The weight loss of whole mushroom during storage was noted. For each weight measurement, the mass of twenty mushrooms was measured. Weight loss was calculated as reported by Nazir et al. [34].

### 2.3. Determinations of Polyphenol Oxidase Activity, Total Phenolic Content, Malondialdehyde (MDA), and Electrolyte Leakage

The extraction and activity of polyphenol oxidase (PPO) were performed as described by Meng et al. [35] with slight modifications. Two grams of frozen tissue was homogenized in 5 mL of phosphate buffer (0.2 M, pH = 6.4) and centrifuged at 12,000× *g* for 15 min at 4 °C. Then, 400 µL of the supernatant was taken and reacted with 2 mL of phosphate buffer and 900 μL 50 mM catechol (Sigma Aldrich, Shanghai, China). Changes in absorbance at 420 nm were recorded over 3 min. A change of 0.01 per minute at 420 nm is defined as one unit (U) of PPO activity. PPO activity was expressed as U per g fresh weight (g^−1^ FW).

Total phenolic content was measured according to the method described by Liu et al. [36]. Half a gram of frozen tissue was homogenized with 5 mL of ice-cold 1% HCl–methanol solution and incubated at 4 °C for 20 min. The mixture was centrifuged at 10,000× *g* for 10 min at 4 °C. The collected supernatant was diluted five-fold with 1% HCl–methanol solution. The absorbance was measured at 280 nm, and the phenolic compounds were expressed as OD_280_ per gram of fresh weight (g^−1^ FW).

Measurement of MDA content was carried out using the method described by Pleșoianu and Nour [37] with some modifications. Two grams of frozen tissue was homogenized with 5 mL of 10% trichloroacetic acid (TCA), and centrifuged at 12,000× *g* for 15 min at 4 °C. One milliliter of the supernatant was reacted with 1 mL of 0.67% 2-thiobarbituric acid (TBA) in a water bath at 100 °C for 20 min, and then cooled immediately to room temperature. The mixture was centrifuged at 12,000× *g* for 10 min. Absorbencies of the supernatant at 450 nm, 532 nm, and 600 nm were measured. The MDA content was calculated using the following formula:MDA content (μmoL g^−1^) = 6.45 × (A_532_ − A_600_) − 0.56 × A_450_.

The determination of electrolyte leakage was conducted according to the previous method proposed by Yan et al. [38]. Mushroom pulp pellets (1 cm in diameter and 0.3 cm in thickness) were placed in a triangular flask with 40 mL double distilled water for 10 min at ambient temperature. The initial conductivity (EC1) of the mixture was measured. The flask was boiled for 10 min and then cooled down to ambient temperature before measuring the final conductivity (EC2). The conductivity of deionized water (EC0) was also measured. Electrolyte leakage was an indicator of the change in membrane permeability and was calculated as follows: Electrolyte leakage (%) = 100% × (EC1 − EC0)/(EC2 − EC0)

### 2.4. Ultrastructural Observation of Cells

The samples were cut into small pieces (1 mm × 1 mm × 3 mm) and immediately pre-fixed in 2.5% (*v*/*v*) glutaraldehyde solution. After rinsing the stationary phase liquid with phosphate buffered saline (PBS), the samples were then post-fixed with 1% osmium tetroxide for 2 h. Subsequently, the samples were cleaned using PBS again and dehydrated in acetone solutions at 30%, 50%, 70%, 85%, and 90% and then twice at 100% acetone. Samples were then placed in SPI 812 epoxy resin embedding agent overnight. Ultrathin sections (70 nm) were cut using an ultramicrotome (EM UC7, Leica Biosystems, Solms, Germany) and observed under transmission electron microscopy (TEM, Hitachi HT7700, Hitachi Ltd., Tokyo, Japan).

### 2.5. Transcriptome Sequencing and Data Analysis

RNA-Seq sequencing libraries were generated by NEB Next Ultra TM RNA Library Prep Kit for Illumina (New England Bio Labs, Ipswich, MA, USA) using high-quality RNA samples and then sequenced on an Illumina platform to generate 150 bp paired-end reads. RNA-Seq was performed using the method described previously [32]. The genome sequence of *A. bisporus* (https://www.ncbi.nlm.nih.gov/datasets/genome/GCF_000300575.1/) was used as the reference (The accessed date is 30 November 2022). Data quality control was performed using fastp v0.19.3 (https://github.com/OpenGene/fastp, accessed on 30 November 2022) to remove reads with adapters. HISAT2 v2.1.0 (http://daehwankimlab.github.io/hisat2/, accessed on 30 November 2022) was used to clean reads by alignment to the reference genome. Gene expression levels were calculated as FPKM (fragments per kilobase million). DESeq2 v1.22.1 (https://bioconductor.org/packages/release/bioc/html/DESeq2.html, accessed on 30 November 2022) was employed to identify the differentially expressed genes (DEGs) with a threshold of the log2 |fold change| ≥ 1 and the false discovery rate (FDR) < 0.05. Three biological replicates were assessed for each sample.

### 2.6. Widely Targeted Metabolomics Profiling and Analysis

The extraction, identification, and quantification of metabolites in *A. bisporus* flesh were conducted by Wuhan Metware Biotechnology Co., Ltd. (Wuhan, China). Briefly, metabolite extracts were analyzed using a UPLC-ESI-MS/MS system (UPLC, ExionLC™ AD; MS, Applied Biosystems 4500 Q TRAP). The sample extracts were injected into a UPLC (Agilent SB-C18, 1.8 μm, 2.1 mm × 100 mm), the mobile phase A was pure water with 0.1% formic acid, and phase B was acetonitrile with 0.1% formic acid. Sample measurements were performed using a gradient program that employed a 95% A, 5% B starting condition. A linear gradient was programmed to 5% A, 95% B over 9 min with a composition of 5% A, 95% B held for 1 min followed by a composition of 95% A, 5.0% B reached in 1.1 min and held for 2.9 min. The flow rate was set to 0.35 mL/min; the column oven was set to 40 °C; and the injection volume was 4 μL. Alternatively, the effluent was connected to an ESI triple quadrupole linear ion trap (QTRAP)-MS.

The ESI source was operated with the following parameters: source temperature 550 °C; ion spray voltage (IS) 5500 V (positive ion mode)/−4500 V (negative ion mode); ion source gas I (GSI), gas II (GSII), curtain gas (CUR) was set at 50, 60, and 25 psi, respectively; the collision-activated dissociation (CAD) was high and scans were obtained in MRM experiments with the collision gas (nitrogen) setting as the medium. The DP (de-clustering potential) and CE (collisional energy) for individual MRM leaps were accomplished by further DP and CE optimization. A specific set of MRM transitions were monitored for each period based on the metabolites eluted during that period. Three biological replicates were performed.

### 2.7. RNA Extraction and Real-Time Quantitative PCR (qPCR)

Total RNA isolation and purification were conducted with Trizol (Takara, Biotechnology, Dalian, China), following the manufacturer’s instructions. Reverse transcription was performed with a HiScript II 1st Strand cDNA Synthesis Kit (Vazyme Biotech, Nanjing, China) according to the manufacturer’s instructions. The relative expression level of each gene was determined by real-time quantitative PCR with HiScript II Q RT Super Mix (Vazyme Biotech, Nanjing, China) and the 2^−ΔΔCt^ method. All the reactions were repeated three times, and the *AbTubulin* gene (AGABI2DRAFT_195658) was used as an internal control for normalization. 

### 2.8. Statistical Analysis

All of the measurements were conducted in triplicate. All statistical analyses were performed by SPSS Statistics 19 (IBM, New York, NY, USA) using one-way analysis of variance (ANOVA). *p* < 0.05 was considered statistically significant for Duncan’s multiple comparison test. The data were fitted and plotted using GraphPad Prism 9.0.0 (GraphPad Software Inc., La Jolla, CA, USA).

## 3. Results

### 3.1. Browning Process of Postharvest A. bisporus Storage at Room Temperature

Variations in the appearance and quality traits of postharvest *A. bisporus* occurred throughout storage. As shown in Figure 1A, comparing with samples before treatment, the mushrooms began to have slight browning symptoms after 2 h of storage and exhibited a sharp increase in browning severity after 6 d of storage (Figure 1A,B), with a browning index reaching 4.43 on 48 h of storage (Figure 1B). We also measured firmness and weight loss at the same time. The firmness of *A. bisporus* showed a decreasing trend during the storage period, with the value declining from 586.21 to 453.37 (Figure 1C), while the weight loss of the mushrooms increased gradually throughout the storage period (Figure 1D).

Browning is an indicator of quality deterioration in *A. bisporus*. PPO catalyzes the oxidative polymerization of polyphenols to form melanin deposits. The total phenolic content and PPO activity of the mushrooms accumulated significantly during the entire storage period (Appendix A). In addition, the increase in PPO activity leveled off between 44 h and 48 h. Cell membrane integrity is very important to ensure the normal operation of the physiological activities of organisms. Here, in our results, the MDA content in *A. bisporus* increased gradually during the first 28 h, and then increased rapidly (Appendix A), and the relative conductivity rate had a similar trend (Appendix A).

### 3.2. Ultrastructural Changes of Cell Membrane Degradation during Ambient Storage of A. bisporus

The ultrastructural changes of the core tissues during storage were observed by transmission electron microscopy. At the initial timepoint (0 h), cell morphology was normal and unperturbed (Figure 2A1–A3). After 24 h of storage, *A. bisporus* cells were smaller, the cytoplasmic membranes were broken, and the mitochondria were degraded. At the same time, the cytoplasmic electron density decreased and cell contents were lost (Figure 2B1–B3). 

### 3.3. Differentially Accumulated Lipid Metabolite Analysis of Flesh under Different Storage Times

To generate a comprehensive insight into the metabolome dynamics in the *A. bisporus* browning process at room temperature for 6 and 24 h, the widely targeted metabolomics technique was carried out to systematically identify and quantify all metabolites present in the mushroom flesh, including primary and secondary metabolites. A principal component analysis (PCA) revealed that the metabolite samples were affected by storage time (Figure 3A). The Pearson’s correlation coefficients (*R*^2^) between biological repeats and between the control and experimental groups were greater than 0.9 (Figure 3B). Altogether, these results indicated that metabolomics data were highly replicable and qualified for further analyses. Further statistics on the metabolites displayed those 893 compounds exclusively grouped into 11 classes, including amino acids and derivatives (301), lipids (162), others (93), nucleotides and derivatives (80), organic acids (75), alkaloids (64), flavonoids (49), phenolic acids (49), terpenoids (13), lignans and coumarins (4), and quinones (3) (Figure 3C; Appendix A). Based on pairwise comparisons, 64 significantly differentially accumulated metabolites (DAMs) (31 up-regulated, 33 down-regulated) were identified between CK and RT_6; 178 DAMs (116 up-regulated, 62 down-regulated) were identified between CK and RT_24; and there were 204 DAMs (149 up-regulated, 55 down-regulated) between RT_6 and RT_24 (Figure 3D). Overall, the number of DAMs increased with the extension of storage time.

To explore the related physiological processes for the DAMs, Kyoto Encyclopedia of Genes and Genomes (KEGG) annotation and analysis were performed. The enrichment analysis showed that a total of seven pathways were significantly affected during the whole RT storage process, including ‘flavone and flavonol biosynthesis’, ‘glycolysis’, ‘purine metabolism’, ‘linoleic acid metabolism’, ‘sphingolipid metabolism’, ‘nucleotide metabolism’, and ‘plant hormone signal transduction’ (*p* < 0.05) (Appendix A). Two out of the seven metabolic pathways identified were closely associated with lipid metabolism, and approximately 18% of metabolites identified in samples were lipids (Figure 3C), indicating that lipids play an important role in the loss of quality during RT storage. Here, we mainly focused on lipid metabolites, and the total of 161 lipid species can be further divided into three classes: phospholipids (PLs, 46.3% total lipid species), fatty acids (FAs, 46.3%), and sphingolipids (SPs, 7.41%) (Figure 4A, Appendix A). Among these lipids, PLs and FAs account for more than 92.6% of the total lipid content, and free fatty acid (FFA, 16:0, 18:0, and 18:1) was the most predominant fraction in FAs (Figure 4A). In *A. bisporus* storage, the vast majority (95%) of PL contents increased at 24 h (Figure 4B), whereas FAs and SPs were differentially accumulated (Figure 4C,D). In general, these results showed that most lipids increased in accumulation during storage at RT.

### 3.4. Transcriptomic Analysis of Flesh under Different Storage Times

To investigate transcriptomic dynamics during RT storage, the transcriptomes at the three storage time points were investigated (Appendix A). Compared with CK, totals of 2859 (CK vs. RT_6), 2966 (CK vs. RT_24), and 1664 (RT_6 vs. RT_24) DEGs were identified (Figure 5A). The number of DEGs between ‘CK vs. RT_6’ and ‘CK vs. RT_24’ was similar, but the combination ‘RT_6 vs. RT_24’ was much less, suggesting that various physiological processes of fruiting bodies were stimulated during 0–6 h storage. The overall expression patterns of up-regulated and down-regulated DEGs among three timepoints were illustrated with Venn diagrams (Figure 5B). Among all pairwise comparisons (‘CK vs. RT_6’ and ‘CK vs. RT_24’ and ‘RT_6 vs. RT_24’), 62 up-regulated genes and 78 down-regulated genes were shared, which indicated that these core genes may be associated with quality maintenance or fruiting bodies’ browning in *A. bisporus*. The DEGs were then classified according to the pathways in which they participate using the KEGG database, with a total of five categories in the first-level classification (Figure 5C, Appendix A). The primary category “metabolism” included the most DEGs. Secondary categories in “metabolism” had more up-regulated DEGs in the early storage period, except for “lipid metabolism” and “energy metabolism”. Notably, in the category of “lipid metabolism”, almost the same up- and down-regulated DEGs were found as in RT_6, but they had higher expression levels in RT_24. These results suggested that lipid metabolism was stimulated during storage. In the primary category “genetic information processing”, the categories of “translation”, “transcription”, and “folding, sorting, and degradation” were dominated by down-regulated DEGs at 6 h, but there were more up-regulated DEGs at 24 h. This indicated that transcription and translation activity was inhibited in RT_6 and significantly activated in RT_24, which may be an adaptation to physiological and metabolic processes. 

As the metabolic analysis revealed that the abundances of major components of fatty acids, phospholipids, and sphingolipids involved in lipid metabolism changed significantly during storage periods, we next identified the DEGs involved in lipid metabolism in the transcriptome and plotted the lipid metabolic pathway (Figure 6). Lipid metabolism includes fatty acid biosynthesis and degradation, galactolipid metabolism, and sphingolipid metabolism. Genes encoding long-chain acyl-CoA synthetase (LACS, AGABI2DRAFT_149127, AGABI2DRAFT_192638, AGABI2DRAFT_195230) associated with fatty acid degradation had higher expression levels in RT_24 compared to those in CK and RT_6. This indicated the degradation of fatty acids might be activated during later storage, and provided abundant precursors for galactolipid and sphingolipid metabolism. Among genes associated with galactolipid metabolism, the genes encoding phospholipase A (PLA, AGABI2DRAFT_151295), phosphatidic acidphosphatase (PAP, AGABI2DRAFT_205847), base-exchange-type phosphatidylserine synthase (PSS, AGABI2DRAFT_224623), phospholipase D (PLD, AGABI2DRAFT_186563), and phosphatidylinositol3-kinase (PI3K, AGABI2DRAFT_151460, AGABI2DRAFT_218336, AGABI2DRAFT_151451) all showed higher transcript level in RT_6, suggesting that galactolipid metabolism was activated during the early period, accompanied by the accumulation of higher levels of lysophosphatidylcholine (LPC) and lysophosphatidylethanolamine (LPE) in RT_24 (Figure 4B). Several genes involved in sphingolipid biosynthesis were down-regulated in RT_6, including those encoding ketosphinganine reductase (KSR, AGABI2DRAFT_186193) and sphingoid base hydroxylase (SBH, AGABI2DRAFT_228680) for phyto-sphinganine biosynthesis, as well as genes encoding enzymes lag 1 homolog (LOH, AGABI2DRAFT_191824), ceramidase (Cdase, AGABI2DRAFT_192931), delta8 sphingolipid long-chain base desaturase (∆8SLD, AGABI2DRAFT_189913), and fatty acid alpha-hydroxylase (FAH, AGABI2DRAFT_194616) for GlcCer biosynthesis. These results suggested that the accumulation of lipids was, to some extent, controlled at the transcription level, thus providing evidence that the lipid metabolism pathway was changed at the transcription level.

### 3.5. Transcription Factors Are Correlated with the Differential Accumulation of Metabolites in A. bisporus

Transcription factors (TFs), especially C_2_H_2_ zinc finger proteins and Zn(II)_2_Cys_6_ transcription factors play key roles in regulating the response of edible fungi to environmental conditions. Heat shock factor (HSF) transcription factors are important regulators of gene expression in response to heat stress in edible fungi. In our dataset, 88 differentially expressed transcription factors (DETFs) belonging to 16 TF families were identified. The most abundant DETFs were members of the C_2_H_2_ (21.59%) and Zn(II)_2_Cys_6_ (20.45%) families, followed by the high mobility group (HMG) (14.77%) and HSF (13.64%) (Figure 7A). Interestingly, most of the C_2_H_2_ (12/19, 63%) and Zn(II)_2_Cys_6_ (12/18, 67%) genes were down-regulated in the flesh at 24 h of ambient storage, although some were up-regulated at 6 h of storage (Figure 7B). These DETFs may contribute to the production of DAMs in the flesh of *A. bisporus*.

### 3.6. Integrated Transcriptomic and Metabolomic Analyses

To further elucidate the changes occurring in *A. bisporus* flesh during ambient storage, we integrated the metabolomics and transcriptomics data. At 6 h of ambient storage, both the DAMs and DEGs in the mushroom flesh were enriched in the same KEGG pathway, which was mainly associated with ‘secondary metabolite biosynthesis and metabolism’, ‘cofactor biosynthesis’, ‘amino acid biosynthesis’, ‘purine metabolism’, and ‘pyruvic acid metabolism’ (Figure 8A). In contrast, after 24 h of ambient storage, the mushroom flesh was associated with additional pathways, including ‘carbon metabolism’, ‘glutathione metabolism’, and ‘proline metabolism’, in addition to the aforementioned pathways (Figure 8B). These findings imply that *A. bisporus* may mitigate the deterioration caused by ambient storage through a series of mechanisms related to amino acid biosynthesis and energy metabolism. 

## 4. Discussion

*A. bisporus* is the most cultivated mushroom world widely [39]. Due to its high respiration and transpiration rates, it is difficult to conserve the quality of harvested button mushrooms, resulting in the loss of taste and high nutritional value [40,41]. *A. bisporus* is very perishable and has a short shelf life of 1–3 days at room temperature (RT) after harvest [42]. Figuring out the physiological and molecular mechanisms behind button mushroom storage at RT is quite helpful for breeding practices and postharvest management. Metabolome and transcriptome data have been combined to comprehensively analyze the physiological and metabolic information of fruits and vegetables; e.g., jujube [43], flowering cabbage [44], radish [31], and citrus [45]. In this study, we conducted a comparative analysis of the metabolome and transcriptome of *A. bisporus* flesh at various time points during ambient storage. We identified numerous differentially abundant metabolites (DAMs) and differentially expressed genes (DEGs). Our findings highlight that the deterioration of *A. bisporus* quality is primarily linked to processes such as membrane lipid metabolism, energy metabolism, and oxidative reactions.

Browning index is the main quality indicator for evaluating the browning of edible mushrooms [46]. In addition, firmness and weight loss are the main factors determining the storage quality of edible fungi, reflecting the degree of softening and quality loss during storage [47,48]. Similar to a previous study [49], button mushrooms stored under 23 °C began to slightly brown after 2 h (Figure 1A,B), with a corresponding decline in firmness (Figure 1C) and weight (Figure 1D). Mushroom browning is a complex process that is regulated by a series of enzymes such as PAL and POD. It is widely believed that PPO oxidizes phenolic substrates into anthraquinone compounds, leading to the occurrence of browning [50]. Here, the content of phenolic substrates showed an increasing trend in ambient storage, unlike previous studies [51,52]. We speculate that the changing trend of phenolic substances is inconsistent due to different storage temperatures of mushrooms and different measurement times. The antioxidant activity of mushrooms is usually related to the total phenolic content. Ambient temperature may stimulate mushrooms to produce more phenolic compounds as a defense response [53]. Moreover, in the present study, the trend of PPO activity was consistent with previous studies in that there was a gradual increase during storage [27,52]. However, in the transcriptomic data, the transcript abundance of *ppo1* and *ppo3* were down-regulated after 6 h of storage at RT temperature and did not change significantly after 24 h. Meanwhile, the transcript abundance of *ppo2*, *ppo4*, *ppo5*, and *ppo6* in mushroom flesh remained unchanged at 6 h and 24 h of storage. These results indicated that PPO-encoding genes might be regulated by multiple factors [27].

The cell membrane system plays an important role in the normal physiological metabolism of fruits and vegetables [54]. MDA is the major product of membrane lipid peroxidation and its accumulation is regarded as an indicator of cell membrane integrity, leading to increased membrane leakage and cellular senescence [55,56]. As previously reported [52], the MDA content and electron leakage of mushrooms increased constantly during the storage, accelerating the senescence of *A. bisporus*. The TEM results further demonstrated that the microstructure integrity of the button mushrooms was disrupted during ambient storage (Figure 2). Additionally, in our metabolome data, the content of phospholipids, sphingolipids, and fatty acids significantly changed, with 95% of the phospholipid content increasing significantly after 24 h of ambient storage (Figure 4). The phospholipids with elevated levels were lysophosphatidylcholine (LPC) and lysophosphatidylethanolamine (LPE), which are the final metabolic breakdown products of phosphatidylcholine (PC) and phosphatidylethanolamine (PE), the most abundant phospholipid components in cell membranes [57,58]. Taken together, the results suggested that the lipid metabolites pathway may be activated in *A. bisporus* during ambient storage. Previous studies have provided evidence that postharvest treatments could reduce MDA content and benefit quality due to the maintenance of cell membrane stability in many different vegetables and fruits, such as pineapples, persimmons, and button mushrooms [59,60,61,62]. The activation of the lipid metabolite pathways in flesh during ambient storage might be a mechanism employed by *A. bisporus* to cope with abiotic stress, such as temperature.

During postharvest storage, a large number of metabolites changed rapidly, causing browning in parallel with the quality deterioration of the button mushrooms. Based on the analysis of the metabolomics, we found a significant increase in the abundances of some major fatty acids, glycerophospholipids, and glycolipids relative to those in CK. Additionally, we characterized differential regulation of key metabolic pathways, including glycolysis, linoleic acid metabolism, sphingolipid metabolism, and nucleotide metabolism (Appendix A). Several strategies in microorganisms indicate that ATP from energy metabolic reactions such as glycolysis and linoleic acid metabolism is involved in the synthesis of fatty acids and phospholipids to repair cell membranes [63,64,65]. We further attempted to establish a regulatory network related to lipid metabolism according to the transcriptomic data (Figure 6). Within this network, transcript abundances of a series of key genes for lipid metabolism, such as *PLD*, *PI3K*, *PSS*, *LACS*, and *IPUT*, were significantly induced during storage This effect likely contributed to increasing the content of fatty acids and sphingolipids. Furthermore, there is evidence that in peach, AP2/ERF superfamily transcription factor ABR1 plays a critical role in regulating lipid metabolism [66,67]. However, the AP2/ERF transcription factor family, is found only in plants [68]. It would be interesting to explore other transcription factors that can regulate lipid metabolism in response to ambient temperature in future studies.

In summary, the metabolic processes of *A. bisporus* related to postharvest quality during ambient storage were investigated using metabolomic and transcriptomic approaches. Changes in lipid metabolites, particularly the high accumulation of LPC and LPE, indicated that the cell membrane structure of *A. bisporus* was dynamically remodeled during ambient storage. The changes may have a great impact on the appearance and texture of *A. bisporus*. Future studies will be focused on developing preservation techniques that inhibit cell membrane degradation. Together, our results provide a theoretical basis and reference for further post-harvest preservation management of *A. bisporus*.

## Figures and Tables

**Figure 1 jof-10-00533-f001:**
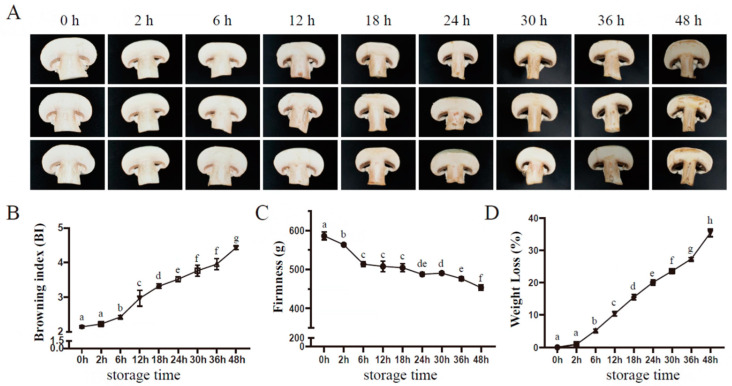
Button mushroom changes in appearance (**A**), browning index (**B**), firmness (**C**), and weight loss (**D**) during postharvest storage at 23 °C. Vertical bars represent standard deviation (*n* = 3). Different lower-case letters represent significant differences between samples under different storage points.

**Figure 2 jof-10-00533-f002:**
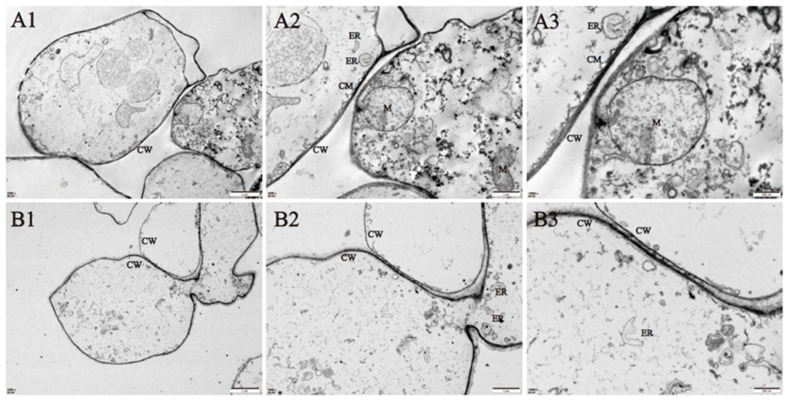
Changes in the cellular ultrastructure of *A. bisporus* before storage at room temperature (**A1**–**A3**) and after 24 h of storage (**B1**–**B3**). CW, cell wall; CM, cell membrane; ER, endoplasmic reticulum; M, mitochondria; Magnification power: (**A1**,**B1**). ×2500; (**A2**,**B2**). ×5000; (**A3**,**B3**). ×10,000. Images are representative of three replicates.

**Figure 3 jof-10-00533-f003:**
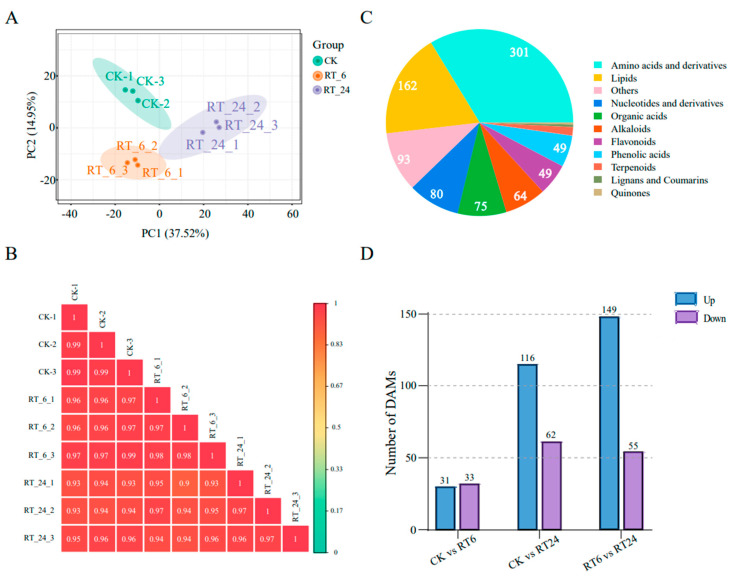
Metabonomic analysis of the button mushrooms under different storage times at 23 °C. (**A**) PCA and (**B**) Pearson’s correlation coefficients of different storage times. (**C**) Pie chart of the types and quantities of metabolites identified. (**D**) The blue and purple bars indicate the numbers of up- and down-regulated DAMs between each comparison group.

**Figure 4 jof-10-00533-f004:**
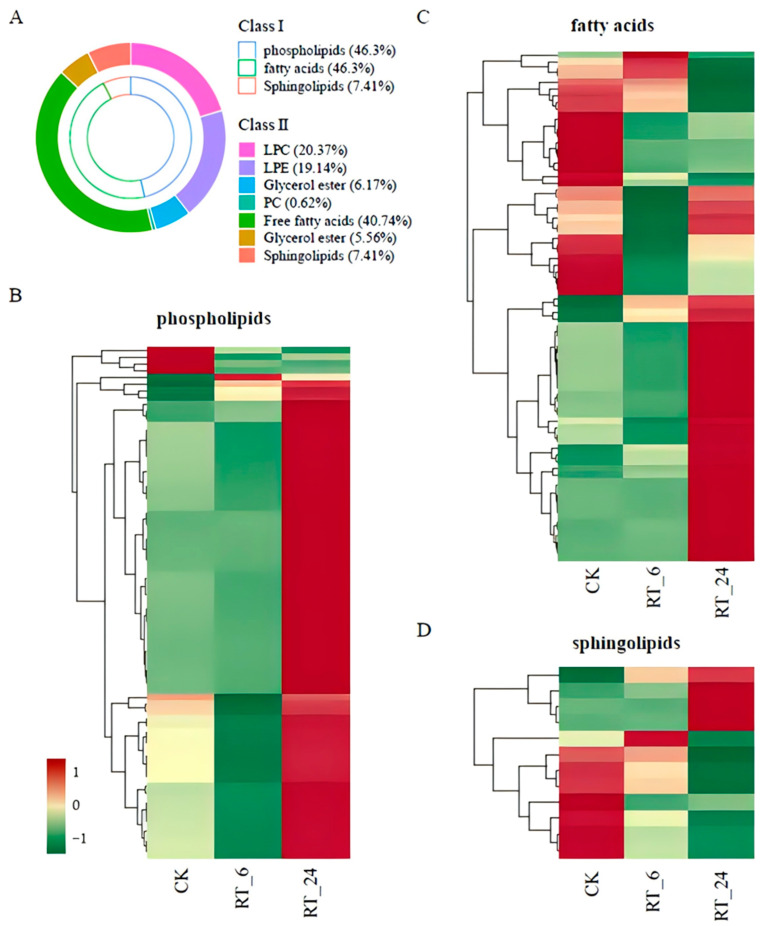
Lipidomic analysis of button mushroom flesh under different storage times at 23 °C. (**A**) The contents of different lipid families and species. Heatmap of contents of phospholipids (**B**), fatty acids (**C**), and sphingolipids (**D**) differentially accumulated in the comparison of CK, RT_6, and RT_24. The red, yellow, and green denote low, middle, and high content. PC, phosphatidylcholine; LPC, lysophosphatidylcholine; LPE, lysophosphatidylethanolamine.

**Figure 5 jof-10-00533-f005:**
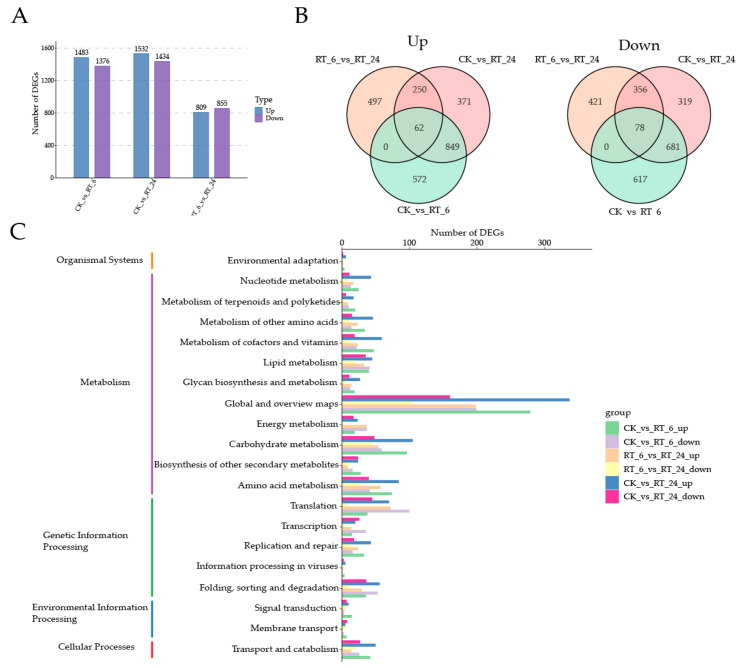
Transcriptomic analysis of button mushroom flesh under different storage times at 23 °C. (**A**) The blue and purple bars indicate the numbers of up- and down-regulated DEGs between each comparison group, respectively. (**B**) Venn diagram of up- and down-regulated differentially expressed genes, respectively. (**C**) KEGG annotation and classification of differentially expressed genes for RT_6 vs. CK, RT_6 vs. RT_24, and CK vs. RT_24. 3.5. Alterations of lipid metabolism pathway genes in the flesh under different storage times.

**Figure 6 jof-10-00533-f006:**
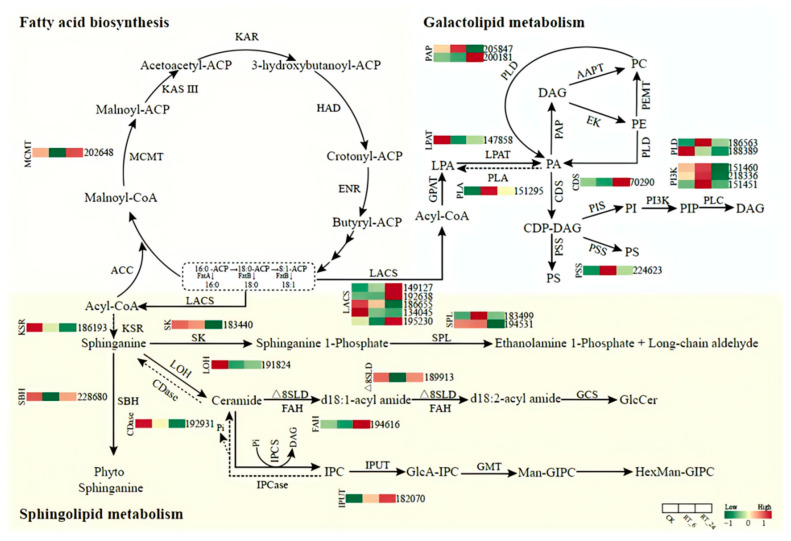
Modulation of lipid metabolism pathway genes during storage at 23 °C. Solid arrows indicate biosynthetic steps and dashed arrows indicate catabolic steps. The expression patterns are presented by heatmap on the basis of log2 FPKM. The color gradient from green to red corresponds to transcript levels from low to high. Abbreviations used are as follows: MCMT, malonyl-CoA: acyl carrier protein malonyltransferase; LACS, long-chain acyl-CoA synthetase; GPAT, glycerol-3-phosphate acyltransferase; LPA, lysophosphatidic acid; LPAT, lysophosphatidic acid acyltransferase; PLA, phospholipase A; PA, phosphatidic acid; PAP, PA phosphatase; CDS, CDP-diacylglycerol synthase; PSS, base-exchange-type phosphatidylserine synthase; PLD, phospholipase D; PI3K, PI 3-kinase; KSR, ketosphinganine reductase; SK, sphingosine kinase; SPL, sphingosine 1-phosphate lyase; LOH, lag1 longevity assurance homolog; SBH, sphingosine base hydroxylase; CDase, ceramidase; ∆8SLD, delta8 sphingolipid long-chain base desaturase; FAH, fatty acid alpha-hydroxylase; IPUT, inositol phosphoryl ceramide glucuronosyltransferase.

**Figure 7 jof-10-00533-f007:**
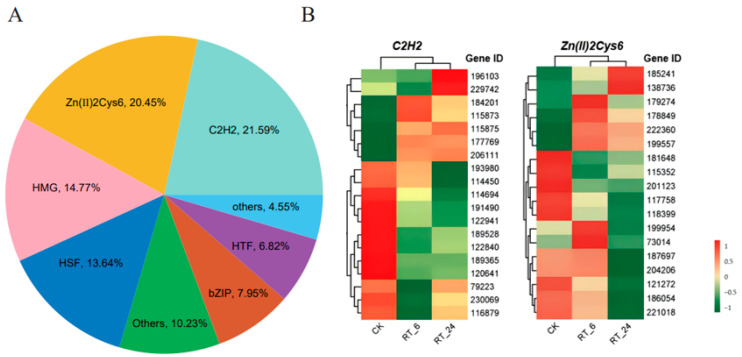
Differentially expressed transcription factors (TFs) in button mushrooms. (**A**) The ratios of differentially expressed transcription factors from different classes. (**B**) Expression profiles of differentially expressed C_2_H_2_ and Zn(II)_2_Cys_6_ family members. Up-regulated (red) and down-regulated (green) genes are indicated.

**Figure 8 jof-10-00533-f008:**
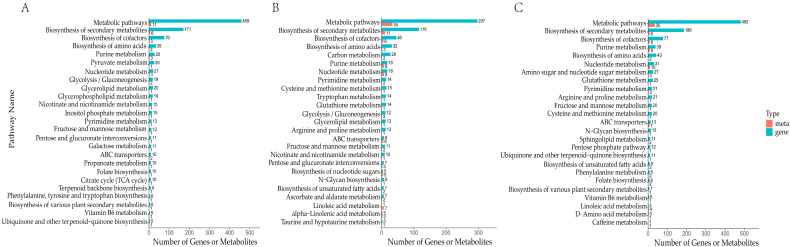
Transcriptome (gene) and metabolome (meta)-combined KEGG enrichment analysis. Joint KEGG enrichment of histograms for RT_6 vs. CK (**A**), RT_24 vs. RT_6 (**B**), and RT_24 vs. CK (**C**).

## Data Availability

All data presented in this study are available. The raw data of RNA-Seq has been registered to the BioProject with a accession number (PRJNA1140854) in the Sequence Read Archive (SRA) database at NCBI (https://www.ncbi.nlm.nih.gov/sra/PRJNA1140854, accessed on 30 November 2022).

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
