# Peer review of "Metabolomic and Transcriptomic Analyses Revealed Lipid Differentiation Mechanisms in Agaricus bisporus at Ambient Conditions"

_jof, 2024, doi:10.3390/jof10080533_

Round 1

Reviewer 1 Report

This is a very interesting article with a great potential for application. All foods spoil during storage and the authors were interested in what happens to mushrooms when they are stored at room temperature. Specifically, metabolomic and transcriptomic approaches were used to show that cell membrane remodeling occurs, accompanied by the accumulation of lysophosphatidylcholine and lysophosphatidylethanolamine.

Major comments

1. In Figure 5C and in Figure 8 a comparison of CK versus RT_24 should also be added.

2. In Figure 8, the up- and downregulated DAMs and DEGs need to be separated.

3. The section “3.2. Lipid metabolites associated ultrastructural changes of cell membrane degradation during ambient storage of A. bisporus” should be divided into 2 parts, related to electron microscopy and metabolomics.

4. It is important to try to identify and discuss which organic compounds are responsible for the browning process.

5. If we compare CK, RT_6 and RT_24 by DEGs, the major gene expression changes occur in the first 6 hours. If we compare them by DAMs, the metabolite changes lag behind the gene expression changes and occur between 6 and 24 hours. Is there any way to show the relationship between the kinetics of these processes more clearly?

6. For what reason are the differences between RT_6 and RT_24 more significant than the differences between CK and RT_24 in Figure 3D? Do the concentrations of some DEGs change first (by 6 hours) and then (by 24 hours) return to the same values? If so, it is very interesting to know which metabolites these are.

7. The conclusions need to be clearer. Perhaps a generalizing picture-schematic diagram of the processes occurring during storage of mushrooms is needed.

1. The first thing that catches your eye is that a large number of words are not separated by spaces. Here are a few examples. Please read the whole text carefully to correct other places.

Str. 12-13, “browningneeds”;

Str. 14, “bisporusat”;

Str. 20, “underlyingl ipidmetabolic”;

Str. 42, “bisporusand”;

Str. 63, “enzymes[24,25].The”;

Str. 99, “2mm”;

Str. 101, “5mm/s”;

Str. 116, “al.[35].”;

Str. 121, “byPleÈ™oi-“;

Str. 131, “al.[37].”;

Str. 152, “generate150”;

Str. 153, “previously[31].The”;

Str. 154, 226, 234, 275, 393 “A.bisporus”;

Str. 155, “readswere”;

Str. 157, “FPKM(fragments”;

Str. 176, “II(GSII)”;

Str. 177, “dissociation(CAD)”;

Str. 187, “instructions.The”;

Str. 199, “processof”;

Str. 204, “1B).We”;

Str. 215, “deposits.The”;

Str. 232, “×2500;A2,B2. ×5000;A3,B3. ×10000”;

Str. 235, “processat”;

Str. 239, “biologicalrepeatsand”;

Str. 261, “lipidmetabolites”;

Str. 265, “flavonolbiosynthesis”;

Str. 271, “onlipid” and “speciescan”;

Str. 272, “phospholipids(PLs”;

Str. 282, “RT_24.The” and “content.PC”;

Str. 294, “withquality”;

Str. 309, “analysisofbutton”;

Str. 320, “encodinglong-chain”;

Str. 396, “RTis”;

Str. 407, “andquality”;

Str. 408, “aprevious”;

Str. 428, “ofcell”;

Str. 433, “ourmetabolome”;

Str. 434, “significantlychanged”;

Str. 445, “bisporusto”;

Str. 460, “ABR1plays”;

Str. 461, “[67].It would”;

Str. 461-462, “interestingto”.

2.      It is also necessary to edit other minor errors and typos, for example:

Str. 177, “QQQ” – should be deciphered or removed;

Str. 242, “sexclusively” – “s” should be removed;

Str. 323, “RT_24” – maybe it was about “RT_6”?;

Str. 457, “PLD, PI3K, PSS, LACS, and IPUT” – gene names should be written in italics.

3.The technical text that is present in MDPI's article form has not been removed from some places. For example, from “The introduction should briefly place...” on str. 26 to “...for further details on references” on str. 34 and “This is a figure. Schemes follow the same formatting” on str. 387.

4.It is necessary to increase the resolution and/or size of all figures, because it is rather difficult to read small captions and the quality of the images is rather poor.

5.The tools used, such as fastpv, HISAT2 and DESeq2, should be cited in section " 2.5. Transcriptome sequencing and data analysis".

Reviewer 2 Report

In this study, metabolomic and transcriptomic data were used to analyze the dynamics of major metabolites and basic regulatory pathways in Agaricus bisporus during different stages of storage at ambient temperature (23 °C). This research confirmed changes in lipid metabolites and a high accumulation of lysophosphatidylcholine and lysophosphatidylethanolamine, products of breakdown of phospholipid components of cell membranes. The authors showed that the structure of the cell membrane of A. bisporus remodeled during storage at ambient temperature. Research significantly contributes to the improvement of post-harvest storage practices.

Suggestions for the manuscript are:

This research may help to elucidate the metabolic mechanisms in A. bisporus 24 hours after harvest. It can also contribute to a better understanding of the influence of ambient temperature on mushroom browning during this period (24h).

Research on molecular mechanisms can provide new insights into postharvest quality optimization.

This research, in comparison with other published materials, points out that the deterioration of the quality of A. bisporus after harvest is associated with the degradation of membrane lipids, as well as with energy metabolism and oxidative reactions.

The main objection to the research is why the introduction did not focus more on the way to store the champignons immediately after picking. Can champignons be stored at a lower temperature after picking? At what temperature are they transported to market chains or factories that pack them in appropriate packages. Also, champignons are sometimes sold unpackaged in markets, i.e. originally displayed on shelves with a lower temperature (from 12-15 C) with proper ventilation. Could this case have served as a control sample?

1. Lines 26-34: The section on instructions to authors should be removed from the introduction.

2. Lines 42-43: Authors should provide a reference for the statement.

Author Response

Reviewer 2

Major comments

In this study, metabolomic and transcriptomic data were used to analyze the dynamics of major metabolites and basic regulatory pathways in Agaricus bisporus during different stages of storage at ambient temperature (23°C). This research confirmed changes in lipid metabolites and a high accumulation of lysophosphatidylcholine and lysophosphatidylethanolamine, products of breakdown of phospholipid components of cell membranes. The authors showed that the structure of the cell membrane of A. bisporus remodeled during storage at ambient temperature. Research significantly contributes to the improvement of post-harvest storage practice.

Answer: Thanks for your positive comments.

Detail comments

Suggestions for the manuscript are:

This research may help to elucidate the metabolic mechanisms in A. bisporus 24 hours after harvest. It can also contribute to a better understanding of the influence of ambient temperature on mushroom browning during this period (24h).

Answer: Thanks for your suggestions. The sampling time points for transcriptome analysis were decided according to the degree of browning of A. bisporus during ambient storage. The three periods of storage were selected: control sampling period (0h), early storage period (6h), and late storage period (24 h). The metabolic changes of A. bisporus at 24 h after harvest are really worth studying. It would take us more time to study the phenolics, polyphenol oxidase and other oxidative enzymes related to A. bisporus browning at 24 h after harvest, and we are doing them and would publish them in another article.

Research on molecular mechanisms can provide new insights into postharvest quality optimization.

This research, in comparison with other published materials, points out that the deterioration of the quality of A. bisporus after harvest is associated with the degradation of membrane lipids, as well as with energy metabolism and oxidative reactions.

The main objection to the research is why the introduction did not focus more on the way to store the champignons immediately after picking. Can champignons be stored at a lower temperature after picking? At what temperature are they transported to market chains or factories that pack them in appropriate packages. Also, champignons are sometimes sold unpackaged in markets, i.e. originally displayed on shelves with a lower temperature (from 12-15 C) with proper ventilation. Could this case have served as a control sample?

Answer: Thanks for your suggestions. In developed countries, energy-intensive factories are used to cultivate mushroom, which is subsequently transported to supermarkets in cold chains after harvest. China, a developing country facing energy constraints, mushroom production often uses low-energy methods. Many mushroom farmers cultivate A. bisporus seasonally in greenhouses, which is similar to the cultivation of the Irish greenhouses. Mushroom is harvested without any pre-harvest cold treatment and are transported directly to the shelves of open-air markets for sale. This storage and transport method result in a very short shelf-life in mushroom production. This cultivation method is less productive than factory-grown mushroom, but it’s still quite profitable because it requires very little of energy. This study is focused on the changes during mushroom storage at ambient conditions to identify the potential improvement practice after post-harvest storage. As there are currently no low-temperature, ventilated boxes for storing mushrooms on shelves available in China. In the mushroom storage experiments, we used the perforated plastic boxes to facilitate ventilation during mushroom storage.

Lines 26-34: The section on instructions to authors should be removed from the introduction.

Answer: Thanks for your comments. We have removed the introductory section in the revised manuscript.

  1. Lines 42-43: Authors should provide a reference for the statement.

Answer: Thanks for your comments. We have added one reference of the current status of A. bisporus production in China in the revised manuscript. Below is the reference information.

Li C, Xu S. Edible mushroom industry in China: current state and perspectives. Appl Microbiol Biotechnol. 2022, 106(11): 3949-3955.

Round 2

Reviewer 1 Report

The authors have responded in detail to all important comments and have greatly improved the manuscript. I have no further significant comments and suggest that the revised manuscript be accepted for publication in the journal.

The authors have responded in detail to all important comments and have greatly improved the manuscript. I have no further significant comments and suggest that the revised manuscript be accepted for publication in the journal.